# WHAT IS IMPORTANT? INTERNAL INTERPRETABILITY OF MODELS PROCESSING DATA WITH INHERENT STRUCTURE

## ABSTRACT

This paper introduces a methodology for constructing interpretable neural networks that quantify the importance of structured input components directly within their internal mechanisms, thereby eliminating the need for traditional explanation methods that rely on post-hoc saliency map generation. Our approach features a two-stage training procedure. First, component specific representations and importance scores are discovered using appropriately designed convolutional neural networks, which are trained jointly. Second, an architecture with relaxed structural constraints, leveraging the previously acquired knowledge, is fine-tuned to capture spatial dependencies among components and to integrate global context. We systematically evaluate our method on Oxford Pets, Stanford Cars, CUB-200, Imagenette, and ImageNet, measuring interpretability-performance trade-offs with metrics for semanticity, sparsity, reproducibility, and, when required, causality (via insertion/deletion-inspired scores). Our architecture achieves improved semantic alignment with ground-truth segmentation annotations compared to post-hoc saliency maps, which, when available, serve as surrogates for expected saliency maps. At the same time, it maintains low variance in importance scores across runs, demonstrating strong reproducibility. Crucially, our architecture provides interpretability gains without sacrificing accuracy. In fact, both with non-pretrained and pretrained backbones, it frequently achieves higher predictive performance than parameter-matched baselines. Overall, compared to both conventional models and post-hoc interpretability techniques under matched computational budgets, our framework produces models that are accurate, stable, and that deliver causally grounded explanations.

## 1 INTRODUCTION

Artificial intelligence is now being deployed across nearly all domains of human activity. Machine learning models are increasingly utilized in decision-making processes that carry critical implications for individuals. However, despite rigorous efforts by model developers to mitigate issues such as overfitting and underfitting, there remains a non-negligible risk that models may exhibit undesirable behavior in real-world deployment. Such behavior may arise due to distribution shifts that are difficult to anticipate, or from latent biases in the training data that lead to the learning of unintended correlations. Consequently, there is an increasing demand for methodologies that enhance the trustworthiness and reliability of machine learning systems. While post-hoc interpretability methods remain popular for analyzing trained models, there is a growing emphasis on designing inherently interpretable architectures - models whose behavior is easier to understand and justify by design (the rationale behind this paradigm shift is elaborated in Appendix A).

In this work, we introduce a methodology that reveals the significance of individual components of a given structure (illustrated with image regions but directly applicable to other modalities such as sequence tokens or graph nodes) within the predictive process. Instead of first modeling complex interactions and then attempting to disentangle them through explanation, our approach begins with a stage dedicated to quantifying component importance in isolation. This design allows each component to be evaluated independently, ensuring that attributions are grounded in the model's internal mechanisms rather than inferred afterward. In a second stage, structural constraints are re-

laxed and models learn spatial and contextual dependencies, while preserving the earlier discovered importance assignments as anchors for interpretability.

## 2 RELATED WORKS

The method discussed in this work assigns attribution scores to elements of the analyzed structure, making it a local explanation technique. In the case of images, this allows for the generation of attribution maps, commonly referred to as saliency maps. The literature offers a wide array of methods for generating such maps. Some techniques, such as LIME Ribeiro et al. (2016), SHAP Lundberg & Lee (2017), and RISE Petsiuk et al. (2018), are perturbation-based and model-agnostic, meaning they can be applied to any model, including black-box ones. In contrast, model-specific approaches leverage the internal characteristics of the analyzed models. Methods like *vanilla gradients*, *guided backprop* Springenberg et al. (2015), *integrated gradients* Sundararajan et al. (2017), and IxG Adebayo et al. (2018) exploit the differentiability of most neural networks. Similarly, LRP Bach et al. (2015) uses gradients to calculate relevance scores. Other methods, such as CAM Zhou et al. (2016), rely on semantic features extracted by CNNs and their specific architecture, while *attention rollout* Abnar & Zuidema (2020) computes the cumulative attention across layers in ViTs. These techniques are often combined, resulting in hybrid methods like GradCAM Selvaraju et al. (2019), GradCAM++ Chattopadhay et al. (2018) and Chefer et al. (2021). From an operational perspective, all of these approaches can be categorized into two groups. Some methods identify which input components influenced the decision, whereas others assess how changes to these components would affect the outcome or compare the current input with a reference to evaluate the impact of their differences. However, despite the large number of attribution methods, concerns remain regarding their reliability. Different techniques may yield conflicting explanations for the same input and model due to varying assumptions. Importantly, all these methods are post-hoc techniques, applied after training. In contrast, our method enforces saliency map generation during inference, allowing the model to use attribution signals directly in its decision-making.

Of course, our method is not the only one aiming to adapt machine learning techniques to facilitate more interpretable behavior. There exists a substantial body of approaches that leverage the high capacity of neural networks, which enables them to solve the same problem in a variety of different ways. This allows for the introduction of additional regularization components into the learning objective, enabling the selection of processing pathways that exhibit desired properties. Some of these approaches are model-agnostic. For example, in Wu et al. (2018), the model is encouraged to mimic the reasoning patterns of a simple decision tree. Other works integrate interpretability constraints directly into the training process, requiring that the explanations derived from the model possess specific qualities, such as fidelity and stability in ExpO Plumb et al. (2020), or adherence to pre-defined priors as in Weinberger et al. (2020). Certain methods exploit structural characteristics of specific models, for instance, Zhang et al. (2018) promotes representations in which the final layers of a convolutional neural network correspond to disentangled, human-understandable concepts. The solution proposed in this work does not require any such complex modifications to the loss function.

Another class of approaches advocates the development of new or modified model architectures. For example, in Böhle et al. (2024), the authors introduce the *B-cos transform* as a substitute for traditional linear transformations. After appropriate training, these transforms become interpretable and align with task-relevant features. A significant subset of these methods involves prototype-based models, inspired by case-based reasoning. In Li et al. (2018), prototype vectors are trained such that object classification can be performed based on distances between the hidden representations of inputs and these prototypes. Additional regularization ensures that the prototypes are meaningful for the task at hand. Since the representations are learned in the latent space of an autoencoder, it is possible, via the decoder, to obtain a semantic interpretation of the learned prototypes. This idea has been further developed in works such as ProtoPNet Chen et al. (2019), PIP-Net Nauta et al. (2023) and ProtoTree Nauta et al. (2021), where convolutional neural networks (CNNs) replace the autoencoder, and prototypes correspond to specific image regions. Similar approaches aiming at the automatic discovery of certain concepts and reasoning based on them can also be found in works using vision transformers (ViTs), where cross-attention is employed for this purpose. Concept Transformer Rigotti et al. (2022) associates inputs with predefined concepts derived from the training data, while INTR Paul et al. (2024) generates concepts across attention heads. In all of these

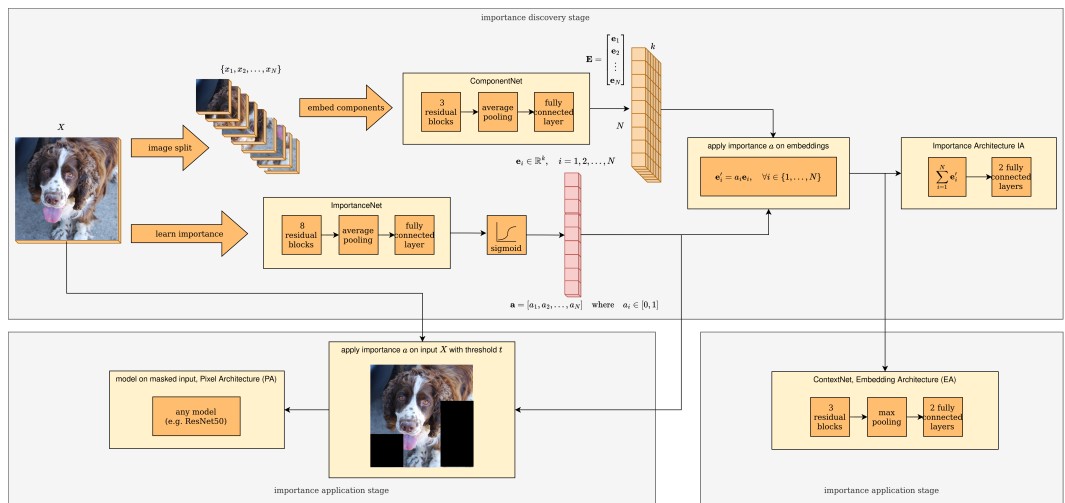

Figure 1: Overview of the proposed method. The input image is processed in parallel to compute patch embeddings and importance scores. The Importance Architecture (IA) identifies informative patches and their relevance. Using these attributions, the Embedding Architecture (EA) applies importance scores to patch embeddings, while the Pixel Architecture (PA) filters the image at the pixel level for improved classification. All three architectures demonstrate how interpretability guides the model to focus on relevant regions.

cases, however, even if image patches are processed as in our method, component representations are computed taking the full image into account, which distinguishes them from our approach.

A conceptually similar approach to the one presented in our paper can be found in several works. Some of these are related to natural language processing. In Hewitt et al. (2023), a language model is built in which each next word is predicted as a weighted sum of multiple embeddings and importance scores computed for every token, reflecting the domain-specific need to capture different token meanings. It employs, however, attention modules, computing token embeddings with access to the entire structure to incorporate inter-token relationships. In Lei et al. (2016), the model identifies text fragments that most strongly support its sentiment analysis decisions. Nevertheless, using a transformer encoder, it again computes component embeddings based on the full sequence context. Additionally, training the importance generator requires extra regularization to enforce sparsity and continuity of the selected fragments. Regarding image processing, two similar approaches can be highlighted. In Wojtas & Chen (2020), the use of dual networks is proposed. The first network (the *operator*) solves the primary task based on masks generated by the second network (the *selector*). Both networks are trained jointly, with the *selector* learning to predict the *operator*'s average output given the masks. However, this setup involves a relatively complex training and inference procedure. In You et al. (2025), importance scores are assigned to automatically discovered groups of patches using attention mechanisms. Both group discovery (the *group generator*) and importance estimation (the *group selector*) rely on attention, with group encodings derived from models accessing entire patch groups - capturing spatial dependencies that we aim to avoid. Moreover, embedding computation and importance estimation are intertwined. Compared to the latter two approaches, our method appears considerably simpler.

In summary, what distinguishes our approach from the related works discussed is that it integrates the problem of learning importance masks directly into the standard training of a single network with a suitably designed architecture. This architectural constraint alone proves sufficient to produce high-quality saliency maps without requiring additional regularization components, which are often essential in many of the methods described. Finally, unlike approaches that first encode complex interactions between components and then attempt to disentangle them through explanation, our method begins with a dedicated stage for independently assessing component-level importance.

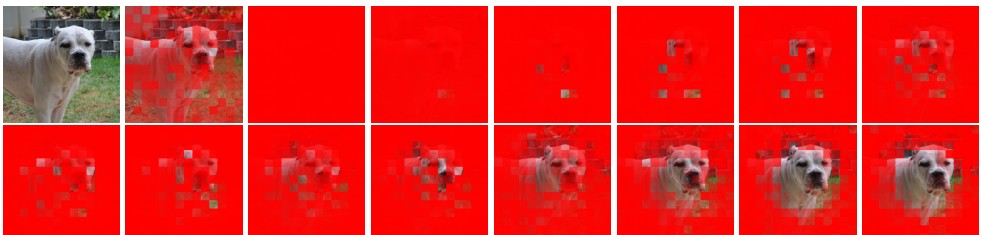

Figure 2: Original image from Oxford Pets dataset and visualization of learning importance vector **a** over epochs of IA(16,128). The not important patches are masked out with red color, the more intense the red color, the less important the patch is.

## 3   METHODS

We propose a methodology for constructing an interpretable neural network that classifies images while explicitly quantifying the contribution of individual input components (image patches). The design separates attribution discovery from contextual modeling. In the first stage, **Importance Architecture (IA)** is employed to address the given classification task. The input image is decomposed into patches, which serve as semantically meaningful units. Each patch is independently processed by a convolutional module to obtain localized embeddings. In parallel, an importance scoring network assigns weights to the patches, quantifying their relative contributions. These two signals are then combined and passed to the classifier. Due to architectural constraints, the classification results at this stage are usually not fully satisfactory. However, in the process of attempting to solve the task as effectively as possible, the model learns to distinguish between information that is useful (important) and information that is irrelevant for task execution. This knowledge can be externally validated to ensure its reliability, and subsequently leveraged in the second stage during the construction of the target model. Free from the aforementioned constraints, most notably by having full access to the spatial relationships between components, the model is capable of achieving state-of-the-art performance while relying solely on the image information previously identified as relevant. In this work, we propose two approaches to constructing such models:

- **Embedding Architecture (EA)** – importance scores are applied directly to the patch embeddings discovered by IA. In this approach, patches are still initially processed independently, which may, in future, enable the discovery of the importance of spatial dependencies among them.

- **Pixel Architecture (PA)** – importance scores discovered by IA are used to filter information directly at the pixel level of the image. This design allows the use of arbitrary, including pretrained, backbones for image analysis, thereby substantially improving classification performance.

Both these architectures also serve to demonstrate that the attribution maps discovered by IA indeed allow for a meaningful distinction between informative and non-informative image regions. All three architectures are illustrated in Figure 1.

### 3.1   IMPORTANCE ARCHITECTURE

The importance architecture (IA) is designed to identify the importance of individual components within an input image, providing an interpretable mechanism for highlighting their relevance to the model's prediction. The input image $\mathbf{X}$ is first divided into $N$ non-overlapping patches, each of size $p \times p$ pixels. These patches are denoted as $\{x_1, x_2, \ldots, x_N\}$, where each patch $x_i$ captures a localized region of the image. Each patch $x_i$ is processed independently by a ComponentNet, consisting of three residual blocks followed by average pooling and a fully connected layer. This maps each patch to a $k$-dimensional embedding vector $\mathbf{e}_i \in \mathbb{R}^k$. The embeddings from all patches are stacked into a matrix $\mathbf{E} \in \mathbb{R}^{N \times k}$, where each row corresponds to the embedding of a specific patch.

To assess the relevance of each patch, the full image $\mathbf{X}$ is also passed through an ImportanceNet, which comprises eight residual blocks, followed by average pooling and a fully connected layer. This produces an importance vector $\mathbf{a} \in \mathbb{R}^N$, which is passed through a sigmoid activation to constrain each value. Consequently, $a_i \in [0, 1]$. These normalized scores are learned during training and represent the relative importance of each patch.

The importance vector $\mathbf{a}$ is applied to the embedding matrix $\mathbf{E}$ through row-wise multiplication, resulting in a weighted embedding matrix $\mathbf{E}' \in \mathbb{R}^{N \times k}$. The rows of $\mathbf{E}'$ are then summed across all patches to produce a single aggregated representation. This summation step is crucial, as it encourages the model to learn semantically meaningful embeddings while focusing on the most relevant patches - effectively masking information that could hinder classification. The aggregated embedding $\mathbf{E}'$ is subsequently passed through two fully connected layers to generate the model's final output. Importantly, these fully connected layers are configured to match the embedding dimensionality, thereby creating a bottleneck that compels ImportanceNet to mask information that is semantically irrelevant and detrimental to the classification process. This approach allows the model, further referred to as IA($p$,$k$), to learn and highlight the most informative parts of the input image, as visualized in Figure 2.

## 3.2 EMBEDDING ARCHITECTURE

While the IA in ImportanceNet effectively captures the relevance of local patches, it lacks the representational capacity to model complex spatial dependencies among them. To address this limitation, we propose an enhanced variant, EA($p$,$k$), which integrates global contextual information and substantially improves overall model performance. In this architecture, the original summation-based aggregation followed by two fully connected layers is replaced by a dedicated ContextNet module. This module processes the weighted patch embeddings $\mathbf{E}'$ using a convolutional neural network composed of three residual blocks, followed by max pooling and two fully connected layers. Importantly, the patch importance weights used to compute $\mathbf{E}'$ are directly inherited from the trained in IA ImportanceNet and are kept frozen during subsequent training. Since ContextNet is relatively lightweight, to further increase the model's representational flexibility, we allow for fine-tuning of ComponentNet. All of that enables the model to better capture intricate visual structures and long-range dependencies across the input image. Moreover, this design choice ensures, an effect that will be demonstrated empirically, that the interpretability afforded by the learned importance scores is preserved.

## 3.3 PIXEL ARCHITECTURE

The pixel architecture PA is designed to explore the interpretability-performance trade-off and to assess the semantic consistency of importance scores by applying them directly to the input images. It uses ImportanceNet with frozen weights from a previously trained IA model to compute the importance scores $\mathbf{a}$, which are then thresholded to retain only the most relevant patches. Specifically, patches with importance values below a predefined threshold $t \in [0, 1]$ are masked out, resulting in a modified (masked) version of the original input. This type of model is denoted as PA($p$,$t$). Note that PA($p$,0.0) corresponds to processing the original, unaltered images (baseline model). The masked input is subsequently processed by a standard backbone architecture, which is trained to make predictions based exclusively on the selected salient regions. By masking the image directly, this approach enforces reliance on the most critical regions, potentially enhancing robustness by filtering out irrelevant visual noise.

## 3.4 TRAINING PROCEDURE

All three architectures were trained using consistent and reproducible configurations. It is important to note that for both EA and PA, the training followed a two-stage procedure, as it was necessary to first train the IA model. Specifically, both IA and EA were trained using the AdamW optimizer with a learning rate of 0.001, a batch size of 128, $\beta_1 = 0.9$, $\beta_2 = 0.999$, and a weight decay of 0.01. Models were trained for 100 epochs on a single NVIDIA RTX 4090 GPU, and the version with the highest validation accuracy was selected for evaluation. Standard data augmentation techniques, such as random resized cropping, horizontal flipping, and normalization, were applied consistently

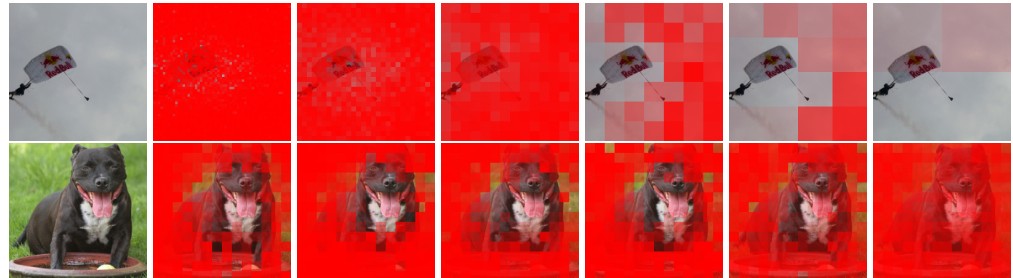

Figure 3: Original images with importance scores obtained using IA($p$,$k$). The first row shows an example from the Imagenette dataset with a fixed embedding size $k = 128$, and varying patch sizes $p \in \{4, 8, 16, 32, 56, 112\}$. The second row shows an example from the Oxford Pets dataset with a fixed patch size $p = 16$, and varying embedding sizes $k \in \{16, 32, 64, 128, 256, 512\}$.

across all training runs. It should be emphasized that, to ensure a fair comparison, the same basic training scheme was employed across all experiments.

## 4    EXPERIMENTS

In our experiments, we aimed to evaluate several key aspects of the proposed approach. First, we assessed semanticity, that is, the extent to which the identified important components align with semantically meaningful content. Second, we introduced a metric to evaluate sparsity, defined as the model's ability to focus exclusively on the truly essential components (the fewer image regions marked as important without degrading classification performance, the better). Third, we examined the reproducibility of the identified important patches across multiple training runs of the same architecture initialized with different random seeds. The results on sparsity and reproducibility are presented in Appendix D and Appendix E, respectively. Moreover, in the case of the EA architecture, we also examined causality (via insertion/deletion-inspired scores), namely whether the employed importance masks indeed influence the model's decisions. This result is included in Appendix F. Finally, we evaluated how restricting predictions to only the most relevant components impacts classification accuracy.

To ensure fair and consistent evaluation, we selected four standard image classification benchmarks. Imagenette Howard (2019) (10 classes) offers easily distinguishable categories for initial validation under low-complexity conditions, with 8500 training, 969 validation, and 3925 test samples. Oxford Pets Parkhi et al. (2012) (37 classes), with well-defined object boundaries, is suitable for assessing semantic structure, and contains 2944 training, 736 validation, and 3669 test images. Stanford Cars Krause et al. (2013) (196 classes) poses a challenge due to high intra-class similarity and subtle inter-class differences, with 7329 training, 815 validation, and 8041 test samples. CUB-200 Wah et al. (2022) (200 classes) is a fine-grained dataset with high intra-class variability, consisting of 4795 training, 1199 validation, and 5794 test images. All the results presented further were obtained for test sets. To systematically explore the influence of architectural scale, we varied the patch size ($p \in 4, 8, 16, 32, 56, 112$) and the embedding dimension ($k \in 16, 32, 64, 128, 256, 512$), and measured their impact across all evaluation metrics and datasets. In addition, we conducted supplementary experiments on the ImageNet Russakovsky et al. (2015) dataset to further assess the scalability of our method. However, due to limited computational resources, the scope of these experiments was restricted. These results are provided in Appendix C. Sample saliency maps for Imagenette are presented in Figure 3. Examples of extracted component importances for all datasets are included in Appendix B.

### 4.1    SEMANTICITY

To evaluate saliency maps, the IoU is commonly used in the literature. It allows for the assessment of the alignment between two binary segmentation masks. However, this metric performs poorly on highly imbalanced images, where background regions dominate. Therefore, we use its macro-averaged version here. Moreover, to further assess the alignment between component importance **a**

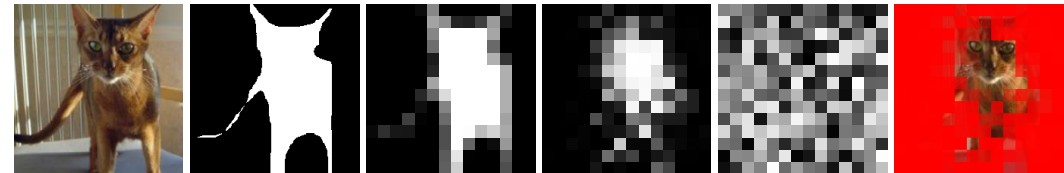

Figure 4: Examples showing: original image, binarized segmentation, averaged patches, learned importance ($d = 0.1127$), random importance ($d = 0.4646$), image with learned mask.

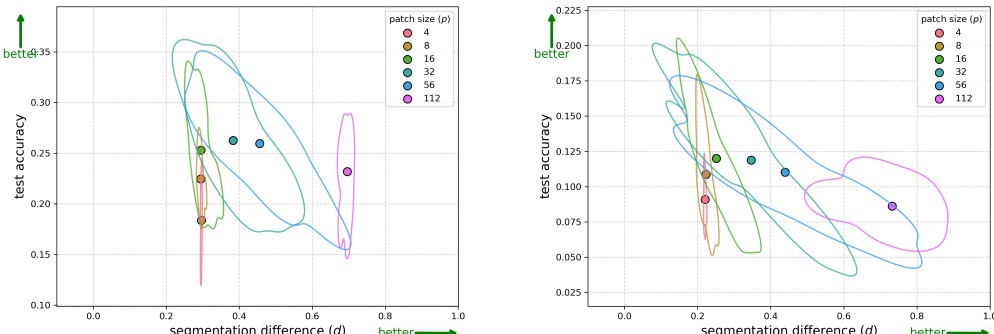

Figure 5: Kernel density estimate plots of accuracy versus segmentation difference for Oxford Pets (left) and CUB-200 (right) from multiple IA experiment runs. Color indicates patch size.

and semantically meaningful regions, we introduce an additional metric, referred to as the segmentation difference $d$:

$$d = \frac{\sum_{j=1}^{T} \sum_{i=1}^{N} \left| m_i^j - a_i^j \right|}{T \cdot N} \tag{1}$$

where $m_i \in [0, 1]$ denotes the patch-wise semantic reference, and $T$ represents the number of samples in the test set. Lower values of the metric indicate a stronger correspondence between the model's importance map and the annotated object boundaries. The variability of this metric across runs and its relationship to predictive accuracy are illustrated in Figure 5.

Among the considered datasets, only the Oxford Pets and CUB-200 datasets provide pixel-level segmentation masks. To compute $m_i$, if necessary, we binarize them by assigning a value of $1$ to foreground objects and $0$ to background pixels, and then average the resulting binary values within each patch. For IoU we binarize the predicted importance scores using a threshold of $0.5$. The procedure used to compute those metric $d$ is illustrated in Figure 4. For methods returning pixel-level saliency maps, we first converted them to patch-level by averaging over $p \times p$ pixel regions, followed by normalization by the maximum value of each map. This ensures consistent scaling across all methods before computing patch-level metrics. In the case of B-cos, we specifically use the alpha channel of the saliency map for this process.

As shown in Table 1, our method achieves the best overall performance. On Oxford Pets, it outperforms all baselines across both metrics, yielding both the lowest segmentation difference $d$ and the highest semantic overlap IoU. On CUB-200, although GradCAM achieves a slightly lower $d$, our method clearly surpasses all others in IoU, indicating a stronger alignment with ground-truth regions. Example visualizations in Appendix B further illustrate these improvements, showing that our learned patch importance produces sharper and more class-relevant explanations compared to classic gradient- or perturbation-based methods.

Table 1: Comparison of learned importance scores across datasets.

| Dataset | Method | PA(16,0.0) pretrained ResNet50 | | PA(16,0.0) non-pretrained ResNet50 | |
|---|---|---|---|---|---|
| | | $d$ | IoU | $d$ | IoU |
| Oxford Pets | GradCAM | 0.3216 | 0.3819 | 0.3873 | 0.4299 |
| | GradCAM++ | 0.3134 | 0.4227 | 0.3696 | **0.4710** |
| | Vanilla gradients | 0.3300 | 0.3626 | 0.3848 | 0.4334 |
| | SHAP | 0.3636 | 0.3071 | 0.4077 | 0.3725 |
| | Occlusion | 0.3789 | 0.2722 | 0.4134 | 0.2720 |
| | Integrated gradients | 0.3523 | 0.3107 | 0.4006 | 0.3780 |
| | Guided backprop | 0.3859 | 0.2636 | 0.4363 | 0.2555 |
| | RISE | 0.3659 | 0.3988 | 0.4032 | 0.4097 |
| | B-cos | 0.3362 | 0.3289 | 0.4234 | 0.3657 |
| | Ours | **0.3065** | **0.4387** | **0.3065** | 0.4387 |
| CUB-200 | GradCAM | 0.1902 | 0.4979 | 0.3905 | 0.4678 |
| | GradCAM++ | **0.1849** | 0.5221 | 0.3819 | 0.4928 |
| | Vanilla gradients | 0.2213 | 0.4656 | 0.3403 | 0.5173 |
| | SHAP | 0.2304 | 0.4176 | 0.3308 | 0.4803 |
| | Occlusion | 0.2284 | 0.3648 | 0.2999 | 0.3991 |
| | Integrated gradients | 0.2180 | 0.4107 | 0.3183 | 0.4568 |
| | Guided backprop | 0.2036 | 0.3869 | 0.2529 | 0.3712 |
| | RISE | 0.2879 | 0.4856 | 0.3582 | 0.4554 |
| | B-cos | 0.2047 | 0.4068 | 0.2997 | 0.4016 |
| | Ours | 0.2280 | **0.5260** | **0.2280** | **0.5260** |

Additionally, for all datasets, semanticity can be assessed indirectly by analyzing the classification results of the EA and PA models, in which irrelevant components of the image are masked either at the level of embeddings or directly on the image, respectively (section 4.2). The results obtained in these settings demonstrate that these models achieve performance that is comparable to, and sometimes better than, the reference model PA($p$,0.0), which processes the entire image. This indicates that the truly relevant parts of the image are indeed identified by ImportanceNet and encoded in the form of importance scores **a**.

## 4.2 ACCURACY

To evaluate the effectiveness of the proposed architectures, we assessed model accuracy across multiple runs. In the case of PA($p$,$t$), this also involved examining how the model performs under different threshold values $t$, which occlude a portion of irrelevant pixels (original, insertion-inspired PA). The best results are reported in Table 2. These findings confirm that our interpretable architectures not only preserve but, in several cases, improve predictive accuracy across all evaluated datasets. Example masked images for the PA model under varying threshold values $t$ are presented in Figure 6.

Furthermore, we investigated how classification accuracy changes with increasing levels of image occlusion (i.e., larger threshold values $t$). The results, shown in Figure 7, further substantiate the conclusion that our interpretable architecture consistently maintains, and in some instances enhances, predictive accuracy. This also suggests that restricting the model's input to the most salient patches, as determined by the importance scores **a**, effectively suppresses noise and mitigates the influence of irrelevant information. For completeness, we also conducted a complementary experiment to confirm that meaningful information is concentrated in patches identified as important, while such information is absent from those deemed unimportant. Specifically, for a given threshold $t$, we occluded patches whose importance scores exceeded this threshold (modified, deletion-inspired PA). The corresponding results are presented in Figure 8 and Figure 9. Information regarding the portion of the image that is masked at different thresholds $t$ can also be found in Tables 4 and 5 in Appendix F.

Table 2: Accuracy comparison of architectures across datasets.

| Dataset | IA($p,k$) | EA($p,k$) | PA(16,0.0) *non-pretrained ResNet50* | PA(16,$t$) |
|---|---|---|---|---|
| Oxford Pets | $31.64 \pm 0.43\%$ ($p = 32, k = 32$) | $48.32\% \pm 1.78\%$ ($p = 32, k = 32$) | $48.30\% \pm 1.82\%$ | $\mathbf{49.86\% \pm 0.28\%}$ ($t = 0.2$) |
| Stanford Cars | $11.50\% \pm 1.10\%$ ($p = 32, k = 64$) | $57.35\% \pm 1.29\%$ ($p = 16, k = 32$) | $62.90\% \pm 1.28\%$ | $\mathbf{65.25\% \pm 1.60\%}$ ($t = 0.1$) |
| Imagenette | $80.14\% \pm 0.09\%$ ($p = 56, k = 64$) | $\mathbf{90.02\% \pm 0.20\%}$ ($p = 56, k = 64$) | $88.67\% \pm 0.31\%$ | $88.09\% \pm 0.94\%$ ($t = 0.1$) |
| CUB-200 | $16.08\% \pm 1.20\%$ ($p = 16, k = 32$) | $40.39\% \pm 0.51\%$ ($p = 16, k = 128$) | $34.69\% \pm 0.77\%$ | $\mathbf{44.06\% \pm 1.04}$ ($t = 0.1$) |

The presented results deviate from state-of-the-art performance, as the models were relatively simple and trained from scratch. Since, in the case of PA, it is possible to employ any backbone, experiments with this architecture were conducted using two pre-trained models: ResNet50 and EfficientNetB4. The results obtained in this setting are, naturally, significantly better, and they are reported in Table 3. In this case, we restricted the evaluation to a single threshold $t$, while also providing information on the amount of occluded, non-essential image area. As can be observed, the results remain highly comparable even when a substantial portion of the image is masked (for $t = 0.0$, the model processes the entire image).

Table 3: Accuracy of PA architecture for pretrained backbones across datasets.

| Dataset | Mask | PA(16,0.0) *pretrained ResNet50* | PA(16,$t$) | PA(16,0.0) *pretrained EfficientNetB4* | PA(16,$t$) |
|---|---|---|---|---|---|
| Oxford Pets | 31.53% | $92.15\% \pm 0.25\%$ | $91.61\% \pm 0.40\%$ ($t = 0.1$) | $92.90\% \pm 0.25\%$ | $92.86\% \pm 0.22\%$ ($t = 0.1$) |
| Stanford Cars | 43.37% | $87.38\% \pm 0.12\%$ | $85.64\% \pm 0.43\%$ ($t = 0.1$) | $87.45\% \pm 0.11\%$ | $86.12\% \pm 0.08\%$ ($t = 0.1$) |
| Imagenette | 44.92% | $97.93\% \pm 0.13\%$ | $92.92\% \pm 0.04\%$ ($t = 0.3$) | $99.62\% \pm 0.08\%$ | $96.61\% \pm 0.21\%$ ($t = 0.3$) |
| CUB-200 | 58.63% | $77.40\% \pm 0.86\%$ | $74.13\% \pm 0.36\%$ ($t = 0.1$) | $79.93\% \pm 0.20\%$ | $77.18\% \pm 0.25\%$ ($t = 0.1$) |

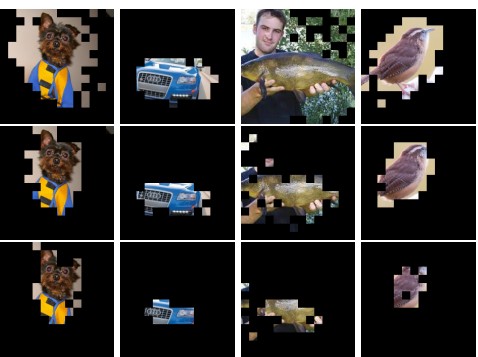

Figure 6: Example masked inputs for each dataset with 0.1, 0.5, 0.9 thresholds for original PA(16,$t$).

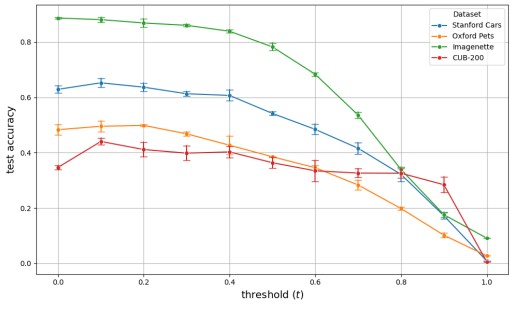

Figure 7: Accuracy at different thresholds for non-pretrained ResNet50 in original PA(16,$t$).

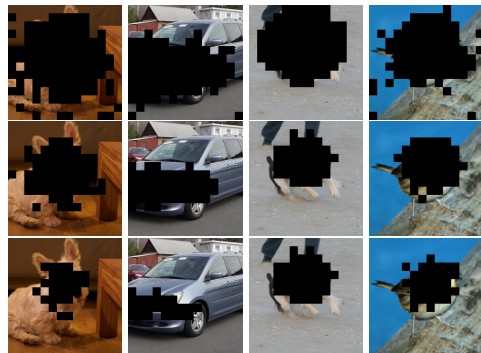 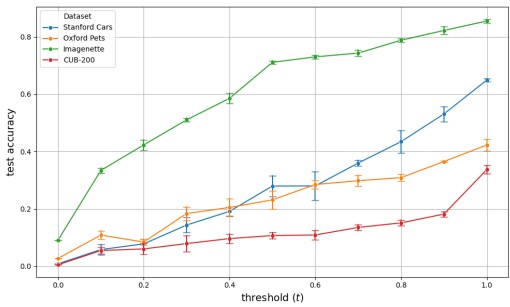

Figure 8: Example masked inputs for each dataset with 0.1, 0.5, 0.9 thresholds for modified PA(16,$t$).

Figure 9: Accuracy at different thresholds for non-pretrained ResNet50 in modified PA(16,$t$).

## 5 CONCLUSION

We introduced a novel interpretable neural architecture that quantifies component importance intrinsically, eliminating the need for traditional post-hoc saliency map generation. Our method employs a two-stage training approach: first learning component-level relevance, then incorporating global context for better prediction. This design results in models that are both highly interpretable and effective, enabling faithful, stable, and semantically meaningful attribution. Experimental results across five benchmark datasets - Oxford Pets, Stanford Cars, Imagenette, CUB-200, and ImageNet demonstrate that our approach achieves strong performance across multiple axes. On Oxford Pets and CUB-200, it achieves the lowest segmentation difference ($d = 0.3065$) and highest semantic alignment ($0.4387$ and $0.5260$, respectively), outperforming standard saliency methods. Visualizations show sharper, class-relevant explanations, while classification accuracy remains high even with masked inputs. These results demonstrate that our model focuses on the most informative regions without sacrificing performance, offering a robust, transparent, and trustworthy framework for interpretable learning.

### REPRODUCIBILITY

To ensure the reproducibility of our results, we provide the complete source code and model weights at the following anonymous link: `https://drive.proton.me/urls/Z09Q9H3K74#UMceSW3CPGnP`. All experiments can be reproduced using these resources, including data preprocessing, training, and evaluation scripts.

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

# A  RATIONALE

Explainable Artificial Intelligence (XAI) aims to address this challenge by providing insights into model behavior tailored to the needs of specific target audiences. Due to the diversity of application domains, model architectures, types of interpretability, and audience requirements, XAI has become a broad and heterogeneous research field encompassing numerous techniques that are difficult to categorize in a unified framework. Nevertheless, it is relatively straightforward to distinguish between two extremes: fully interpretable models and black-box models. The former assume full transparency regarding model internals, with inherent simplicity that allows human users to understand the decision-making process. Examples include various forms of linear models, case-based inference and decision trees or rule-based systems. Given their transparent structure, it is generally feasible to trace the reasoning process and identify which input features are most influential in determining the output. The main limitation in such cases arises when the input space is too high-dimensional or semantically complex for humans to interpret individual features meaningfully. In contrast, black-box models either offer no access to their internal mechanisms or are so complex that their internal logic is deemed intractable, even if technically accessible. For these models, two primary forms of post-hoc explanations are typically employed: local explanations, which aim to interpret individual model predictions, and global explanations, which attempt to characterize the model's overall behavior. Both approaches frequently rely on perturbation-based techniques, which analyze the model's output under systematically varied inputs. Alternatively, surrogate models, i.e. simple interpretable models trained to approximate the behavior of the original model, may be used. However, surrogate-based explanations tend to be reliable only in local regions of the input space, and even then, the surrogate's reasoning process may diverge significantly from that of the original model, potentially leading to misleading interpretations. As a result, the most commonly used strategies for establishing trust in black-box models are attribution methods (discussed in Section 2) and counterfactual explanations. Attribution methods estimate the contribution of individual input features to a specific prediction, while counterfactuals identify minimally modified inputs that would result in different model outputs. These explanation outputs are evaluated by the intended users and compared against domain knowledge or expectations to assess the model's validity. Naturally, the effectiveness of both approaches is also constrained by the complexity of the input space, which may limit interpretability.

Fully interpretable models are typically relatively simple and tend to underperform on complex tasks. Moreover, there is growing criticism in the literature concerning the validity of explanations for black-box models, including a lack of trust in the resulting attributions Rudin (2019); Lakkaraju & Bastani (2020); Laugel et al. (2019). Fortunately, between the two extremes outlined above, there exists a broad spectrum of intermediate approaches. These approaches, which have at least partial access to model internals, can offer additional cues that facilitate the explanation process. Such cues may arise naturally from built-in mechanisms inherent to the model architecture. For instance, convolutional neural networks (CNNs) were originally designed to emulate certain functional principles of the human visual system and are thus capable of progressively extracting semantic features across layers. When appropriately disentangled, these intermediate representations can be used to derive pixel-level attributions. Similarly, in vision transformers (ViTs), attention coefficients are explicitly designed to capture the relative importance of different input components and can be leveraged for interpretability purposes in an analogous way. However, those cues may be also deliberately incorporated by model designers (as discussed in Section 2), which is done in this work.

# B  EXAMPLES

This section presents qualitative examples illustrating the behavior of our attribution methods and the masks generated from the learned importance maps. We first compare attribution maps produced by various interpretability techniques on both pretrained and non-pretrained models (Figures 10 and 11). We then show original images alongside their masked counterparts across multiple datasets - Stanford Cars, Oxford Pets, Imagenette, CUB-200, and ImageNet (Figures 12–16). These examples demonstrate how our IA models highlight informative regions and how the resulting masks vary across datasets and model configurations.

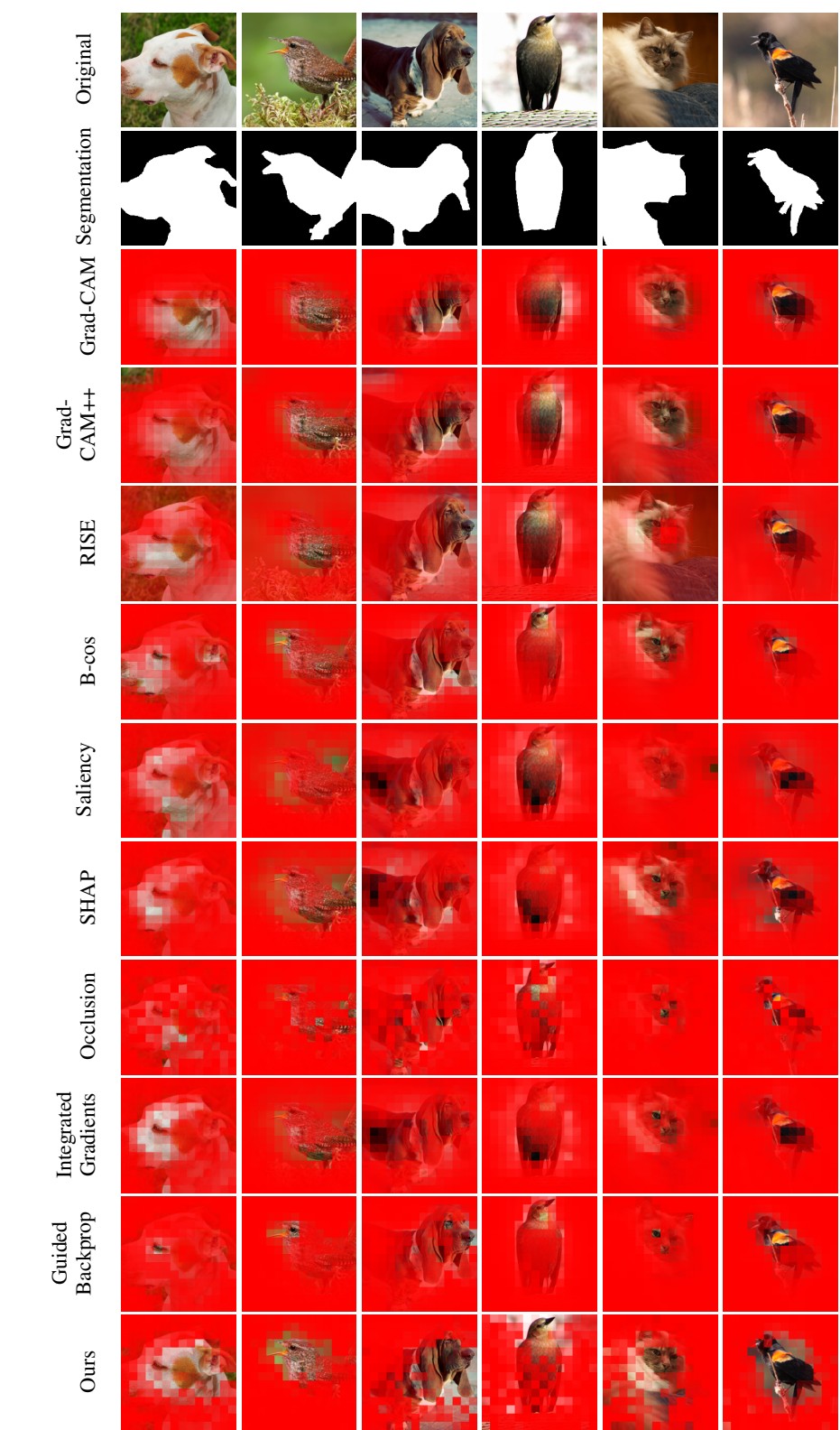

Figure 10: Visualizations of attribution maps for different methods on Oxford Pets and CUB-200 (pretrained model).

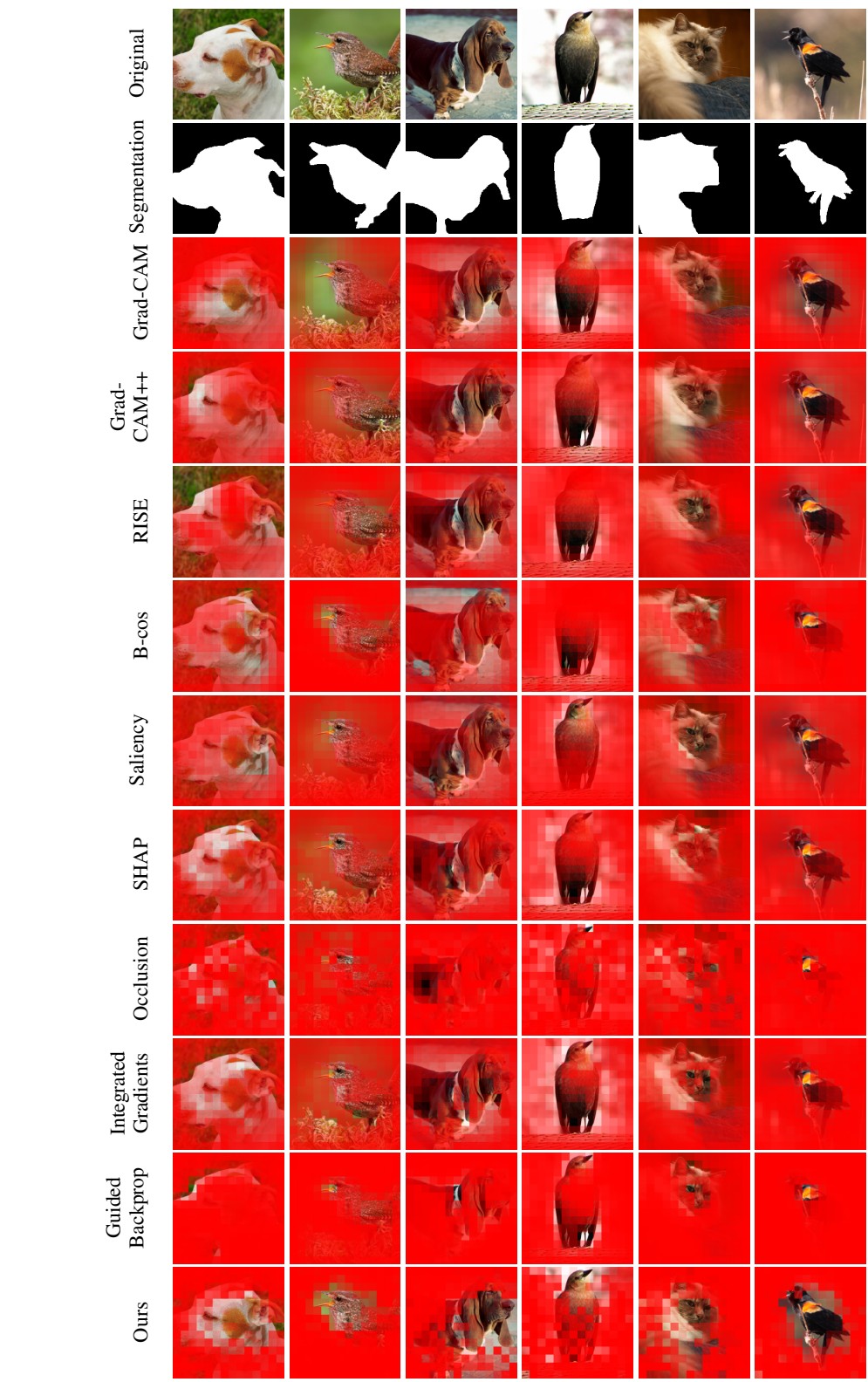

Figure 11: Visualizations of attribution maps for different methods on Oxford Pets and CUB-200 (non-pretrained model).

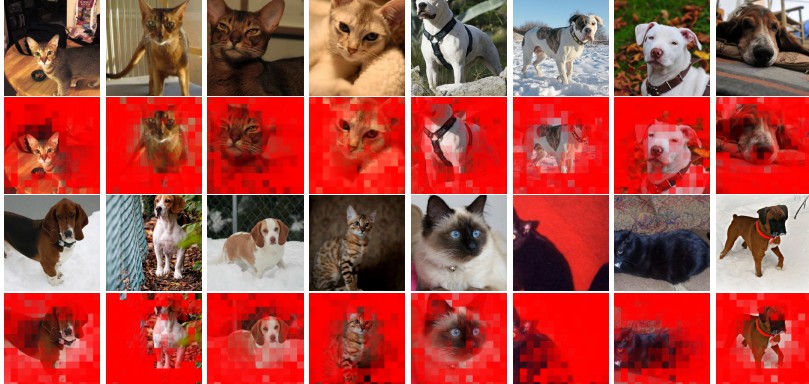

Figure 12: Original and masked images from Oxford Pets dataset. The first row shows the original images, while the second row displays the corresponding masked images. The masks are generated using the importance maps obtained from our IA(16,128) model.

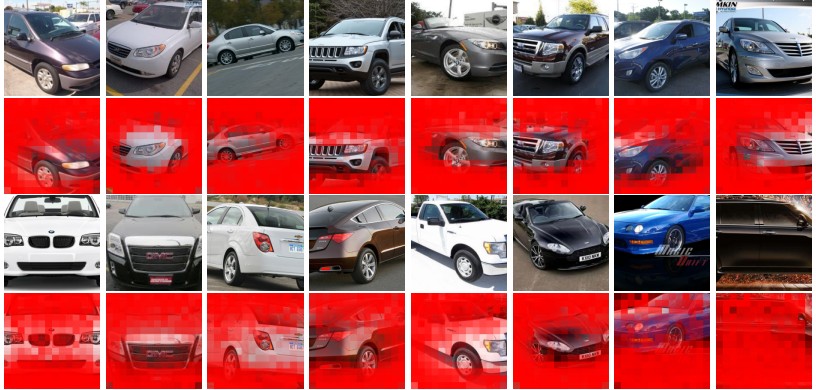

Figure 13: Original and masked images from Stanford Cars dataset. The first row shows the original images, while the second row displays the corresponding masked images. The masks are generated using the importance maps obtained from our IA(16,128) model.

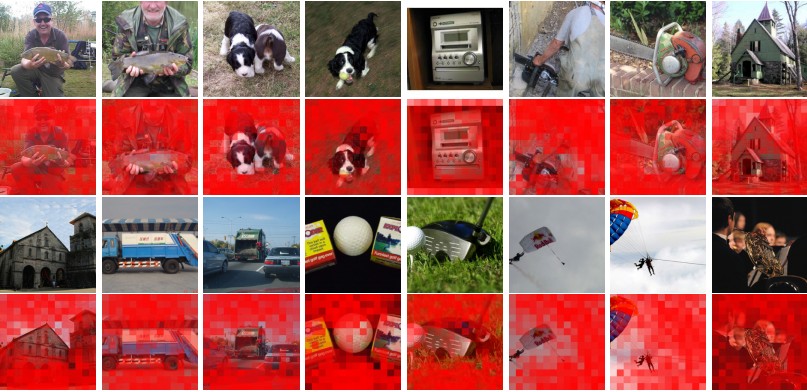

Figure 14: Original and masked images from Imagenette dataset. The first row shows the original images, while the second row displays the corresponding masked images. The masks are generated using the importance maps obtained from our IA(16,128) model.

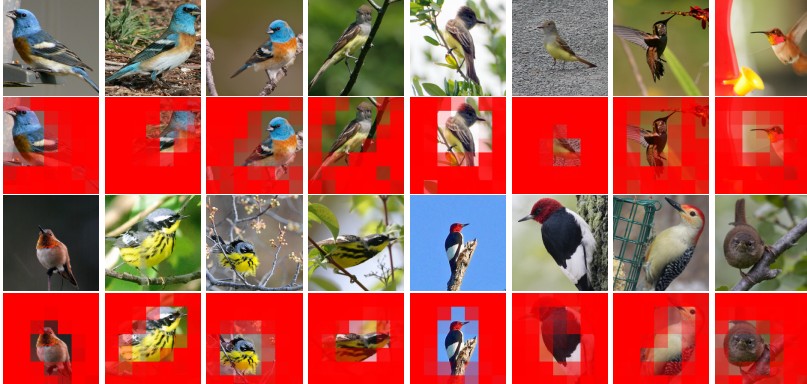

Figure 15: Original and masked images from CUB-200 dataset. The first row shows the original images, while the second row displays the corresponding masked images. The masks are generated using the importance maps obtained from our IA(32,32) model.

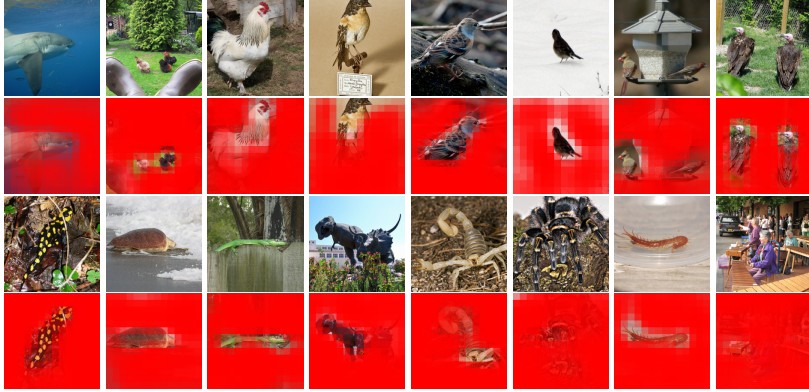

Figure 16: Original and masked images from ImageNet dataset. The first row shows the original images, while the second row displays the corresponding masked images. The masks are generated using the importance maps obtained from our IA(16,128) model.

## C SCALABILITY

To assess the scalability of our method, we conducted experiments on the challenging ImageNet dataset, using a model configuration with $p = 16$ and $k = 128$, chosen to balance computational feasibility with representational capacity. The results, shown in Figure 17, reveal that our learned importance maps consistently focus on semantically meaningful regions. These masks concentrate on primary objects suppressing irrelevant background details. This behavior demonstrates that the model effectively acquires a notion of visual importance aligned with human perception.

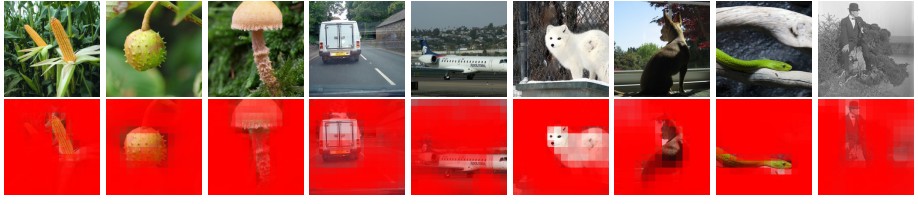

Figure 17: Original and masked images from ImageNet dataset for IA(16,128)

To evaluate how much input information can be removed without harming performance, we introduced sparsity by thresholding the importance values. A threshold of $t = 0.01$ was selected, which on average masks 17% of each image in the ImageNet test set. As illustrated in Figure 18, the thresholded images retain their structural and semantic clarity, suggesting that a small subset of highly informative pixels is sufficient to preserve task-relevant content.

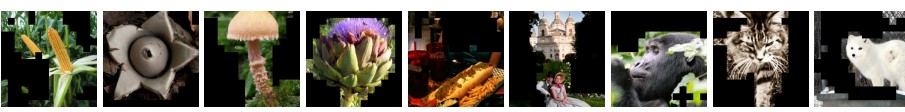

Figure 18: Images from ImageNet dataset for PA(16,0.01).

Performance results further highlight the significance of architectural choices. The IA(16,128) model, which learns solely from feature of local components without access to any global context, achieved accuracy of 37.35%. This result demonstrates that while localized importance cues are informative, they are insufficient on their own for high-level visual understanding. To address this limitation, we trained EA(16,128), which augments the model with a global information. This version markedly improves performance, reaching accuracy of 55.78%, confirming that even limited contextual modeling enhances the system's ability to interpret images more holistically. It is worth noting that the EA model remains shallow and compact, and its performance could be further boosted by increasing its depth or employing more sophisticated architectural components.

The most compelling outcome comes from the PA models, which incorporates both global context and refined attention, operating on the thresholded input at $t = 0.01$. Despite discarding a substantial portion of each image, this configuration achieves accuracy of 75.42%, matching that of a standard ResNet50 baseline equivalent with PA(16,0.0) and coming close to the ViT-B/16 Dosovitskiy et al. (2020), which achieved 77.91% with a higher image resolution and more than 3 times the number of parameters. This outcome is particularly noteworthy because it confirms that the PA model not only captures the essential semantic content of the images, but does so efficiently and sparsely, without relying on the full input signal. The PA results thus serve as strong evidence that the importance maps generated by our method are both meaningful and actionable even for large scale datasets.

## D SPARSITY

Sparsity quantifies how selectively the model utilizes image patches, directly influencing interpretability. A lower sparsity score indicates that the model focuses on fewer, more relevant patches, which is desirable for interpretability. We define the sparsity metric as the average importance scores across all patches and images:

$$s = \frac{\sum_{j=1}^{T} \sum_{i=1}^{N} a_i^j}{T \cdot N} \tag{2}$$

Figure 19 illustrates how sparsity varies across datasets and patch sizes, revealing a clear inverse relationship: as patch size increases, sparsity decreases. Smaller patches tend to yield higher sparsity scores, suggesting that the model is more selective and focuses on finer image details. In contrast, larger patches result in lower sparsity because the model distributes importance across fewer, broader regions, reducing the need for selective focus. Additionally, this metric is valuable because lower sparsity values show that the model uses only a few important features, making it easier to distinguish between relevant and irrelevant regions. This focused attention improves interpretability and helps the model generalize better.

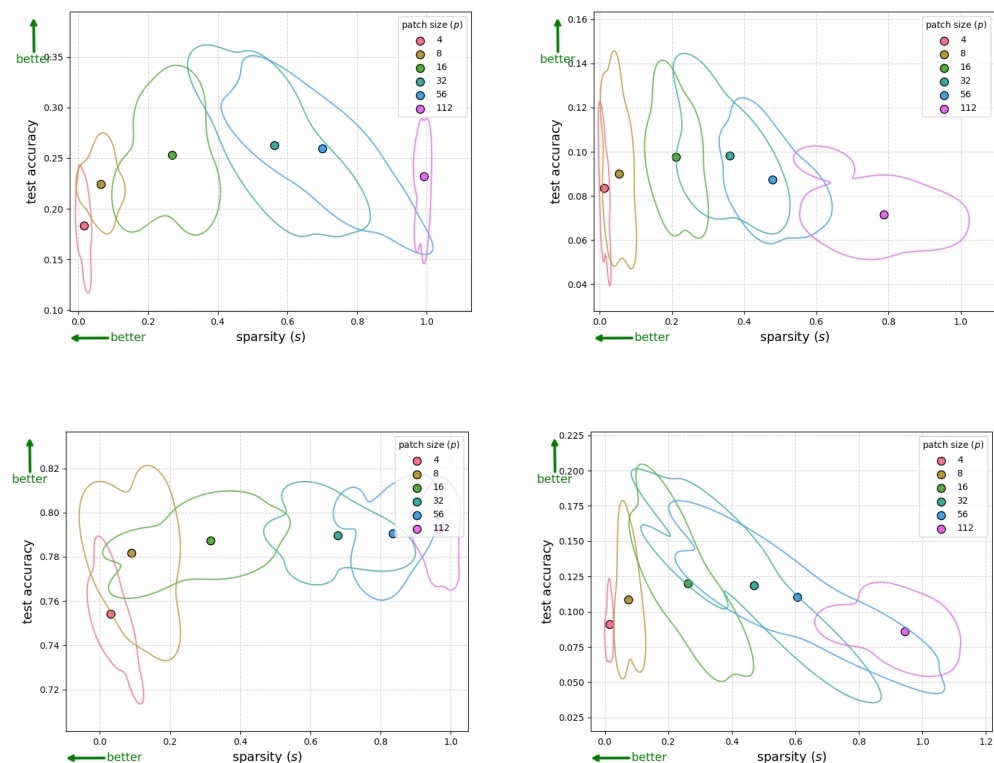

Figure 19: Kernel density estimate plots of accuracy versus sparsity for Oxford Pets, Stanford Cars, Imagenette and CUB-200 datasets from multiple IA experiment runs with color indicating patch size.

## E    REPRODUCIBILITY

Reproducibility quantifies model stability across random initializations by measuring standard deviation of importance scores. For each test sample $j$ and patch $i$, we compute the standard deviation $SD$ of its importance scores $a_i^j$ and define reproducibility metric as:

$$r = \frac{\sum_{j=1}^{T} \sum_{i=1}^{N} SD(a_i^j)}{T \cdot N} \tag{3}$$

Lower values indicate higher reproducibility - model importance maps remain stable despite random initialization changes. This metric reveals whether learned explanations reflect consistent patterns versus seed-dependent artifacts. The example values of reproducibility are shown in Figure 20 and Figure 21.

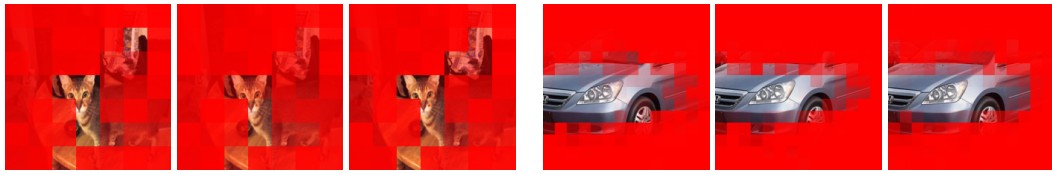

Figure 20: Example of reproducibility from Oxford Pets IA(32,128) with standard deviation $r = 0.06$

Figure 21: Example of reproducibility from Stanford Cars IA(16,128) with standard deviation $r = 0.02$

Results presented on Figure 22 across different embedding sizes and patch sizes demonstrate low standard deviation, which suggest that the model is able to learn stable importance maps. We see a particularly low standard deviation (under 0.1) for small patch sizes, suggesting that the model's reproducibility is more stable and consistent when finer-grained spatial information is preserved, in the case of larger patches, the deviation is higher due to the many elements contained in one patch. It is also worth noting that larger embedding sizes (e.g., 512) tend to result in a smaller standard deviation, suggesting a more stable training process.

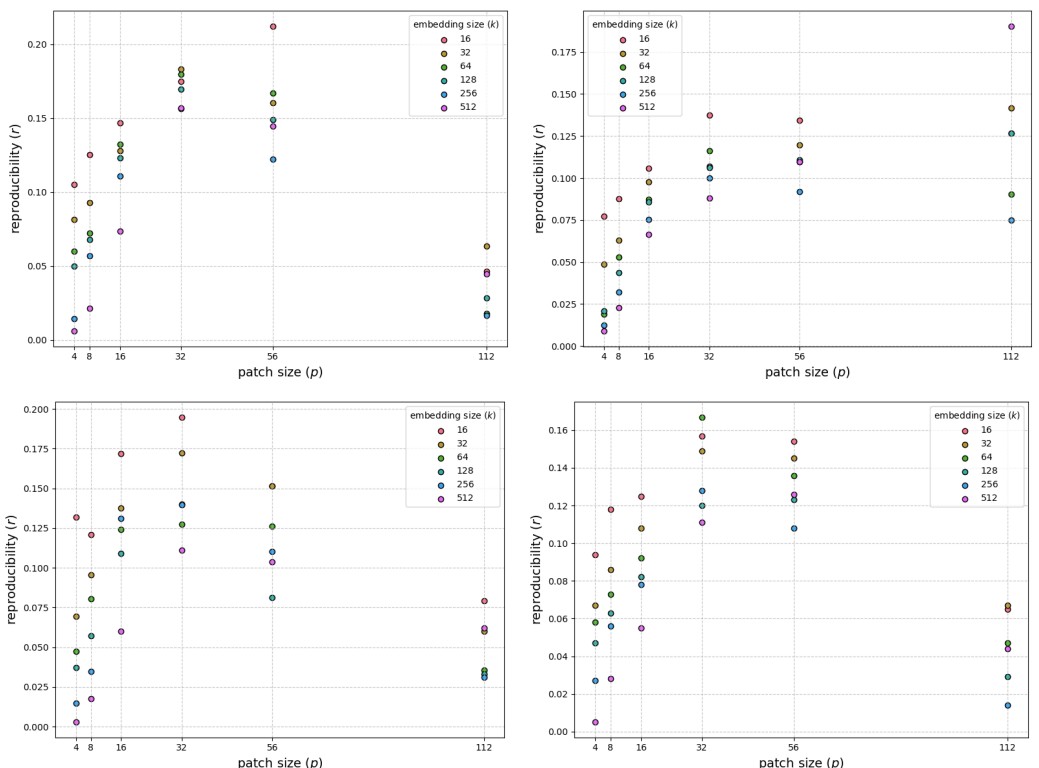

Figure 22: Reproducibility of importance scores across different patch sizes and embedding sizes for Oxford Pets, Stanford Cars, Imagenette and CUB-200 datasets.

## F CAUSALITY

In the case of the EA architecture, there exists a possibility that the network might ignore the patch-importance information and, by appropriately adjusting the weights of ComponentNet and ContextNet, recover the masked content. To test whether this model genuinely relies on the importance scores produced by the IA, we conducted insertion/deletion-style experiments. While there are several differences between the proposed approach and the original definition of the insertion/deletion metrics introduced in Petsiuk et al. (2018) and further generalized in Covert et al. (2021), the overall idea and the ability to assess the causality of the evaluated importance masks are preserved. These differences include:

- Instead of zeroing out pixel values, we remove (zero) the embeddings. However, since the same ComponentNet module processes all patches, zeroing pixels would effectively also erase the information contained in the corresponding embeddings.
- For the insertion score, the standard procedure adds regions starting from the most important. We simply reverse the order and remove regions beginning with the least important. Similarly, for the deletion score, rather than removing regions from the most important, we reveal the least important ones. This modification only alters the order of value computation without changing the underlying metric.

- In the original method, a region of fixed size is added or removed at each step, which can influence the amount of information revealed or hidden depending on object sizes. To mitigate this, we reveal or hide patches whose importance scores exceed successive thresholds $t$ at each step.

- To allow comparison across individual images and entire datasets, the original approach condensed the step-wise model performance into a single number via AUC computation. We retain the raw values instead, analyzing them directly.

- In the original work, model performance was measured either by the probability of either predicted or expected label. To avoid this choice and provide a holistic assessment of the model's response, we simply measure whether each prediction is correct. Averaging these values over the dataset corresponds to computing step-wise accuracy.

In summary, for each patch $i$, the learned importance score $a_i$ was compared against a threshold $t$. In the insertion setting, all patches with $a_i < t$ were zeroed, ensuring that the EA operated without receiving semantic information from patches deemed irrelevant by the IA. In the deletion setting, all patches were initially removed and then progressively restored in order of increasing importance. Consequently, all patches with $a_i > t$ were zeroed, and the model operated solely on patches considered irrelevant.

Table 4: Effect of removing low-importance patches on accuracy (insertion-inspired evaluation) for EA(16,128).

| $t$ | Imagenette | | Oxford Pets | | Stanford Cars | | CUB-200 | |
|---|---|---|---|---|---|---|---|---|
| | Acc (%) | Mask (%) | Acc (%) | Mask (%) | Acc (%) | Mask (%) | Acc (%) | Mask (%) |
| 0.0 | 88.38 | 0.00 | 49.99 | 0.00 | 55.91 | 0.00 | 39.92 | 0.00 |
| 0.1 | 88.38 | 4.87 | 49.88 | 31.53 | 54.94 | 43.37 | 39.02 | 58.63 |
| 0.2 | 87.92 | 21.12 | 49.33 | 46.31 | 53.08 | 58.28 | 37.76 | 71.53 |
| 0.3 | 85.61 | 44.92 | 48.32 | 56.90 | 50.14 | 68.03 | 35.88 | 78.73 |
| 0.4 | 80.33 | 67.34 | 47.40 | 65.52 | 46.05 | 75.13 | 33.21 | 83.59 |
| 0.5 | 69.25 | 82.74 | 45.27 | 72.86 | 40.82 | 80.73 | 30.14 | 87.28 |
| 0.6 | 51.49 | 91.83 | 41.84 | 79.46 | 34.68 | 85.39 | 27.13 | 90.27 |
| 0.7 | 34.01 | 96.66 | 37.48 | 85.23 | 27.27 | 89.50 | 23.54 | 92.82 |
| 0.8 | 20.15 | 98.98 | 29.98 | 90.55 | 19.04 | 93.30 | 18.05 | 95.14 |
| 0.9 | 11.75 | 99.86 | 19.43 | 95.59 | 9.05 | 96.97 | 11.87 | 97.44 |
| 1.0 | 9.94 | 100.00 | 2.73 | 100.00 | 0.52 | 100.00 | 0.50 | 100.00 |

Table 5: Effect of adding low-importance patches on accuracy (deletion-inspired evaluation) EA(16,128).

| $t$ | Imagenette | | Oxford Pets | | Stanford Cars | | CUB-200 | |
|---|---|---|---|---|---|---|---|---|
| | Acc (%) | Mask (%) | Acc (%) | Mask (%) | Acc (%) | Mask (%) | Acc (%) | Mask (%) |
| 0.0 | 9.94 | 100.00 | 2.73 | 100.00 | 0.52 | 100.00 | 0.50 | 100.00 |
| 0.1 | 13.17 | 95.13 | 6.56 | 68.47 | 2.50 | 56.63 | 4.40 | 41.37 |
| 0.2 | 36.51 | 78.88 | 12.79 | 53.69 | 6.37 | 41.72 | 7.66 | 28.47 |
| 0.3 | 64.74 | 55.08 | 17.90 | 43.10 | 10.08 | 31.97 | 10.98 | 21.27 |
| 0.4 | 78.24 | 32.66 | 22.30 | 34.48 | 15.73 | 24.87 | 14.52 | 16.41 |
| 0.5 | 83.34 | 17.26 | 26.13 | 27.14 | 20.95 | 19.27 | 17.38 | 12.72 |
| 0.6 | 86.19 | 8.17 | 30.25 | 20.54 | 26.44 | 14.61 | 20.59 | 9.73 |
| 0.7 | 86.98 | 3.34 | 33.87 | 14.77 | 33.38 | 10.50 | 24.11 | 7.18 |
| 0.8 | 87.52 | 1.02 | 37.57 | 9.45 | 42.04 | 6.70 | 28.24 | 4.86 |
| 0.9 | 88.35 | 0.14 | 43.39 | 4.41 | 52.63 | 3.03 | 32.55 | 2.56 |
| 1.0 | 88.38 | 0.00 | 49.99 | 0.00 | 55.91 | 0.00 | 39.92 | 0.00 |

The results are summarized in Tables 4 and 5. Insertion-inspired experiments show that removing low importance patches leads to only slight drops in accuracy until a large fraction of patches is discarded (e.g., Imagenette remains at $87.9\%$ accuracy even when $21\%$ of patches are masked). Deletion-inspired experiments, in contrast, start from an empty image and gradually recover performance as patches are reintroduced in order of increasing importance. Adding the least important patches produces little to no accuracy gain (e.g., Imagenette remains below $20\%$ even after $30\%$ of patches are restored), while accuracy only rises sharply once the most important patches are included. For Imagenette, performance eventually reaches $88.35\%$ when less than $1\%$ of patches remain masked, essentially matching the baseline. Similar recovery dynamics are observed across Oxford Pets, Stanford Cars, and CUB-200. These findings confirm that the importance masks are

causally linked to decision-making: irrelevant patches can be discarded with minimal cost, while accurate predictions emerge only when the patches identified as important by IA are restored.

## G   LIMITATIONS

Our experiments on Oxford Pets, Stanford Cars, Imagenette, CUB-200 and ImageNet show that the proposed architecture achieves strong interpretability, semantic alignment, and predictive performance. The learned importance maps are reproducible, semantically meaningful, and often improve classification. However, certain design aspects suggest opportunities for refinement. In binary classification tasks, ImportanceNet can produce overly polarized maps, with values clustered near $0.0$ or $1.0$ reflecting confident attribution but potentially limiting interpretability by obscuring intra-class variation. This is more evident in low-complexity tasks, where the model's capacity may be excessive. More compact ImporanceNet variants could address this by preserving finer-grained attributions without sacrificing performance. Additionally, our second-stage module, ContextNet, effectively captures global context but is limited in flexibility. Replacing it with transformer-based attention or graph neural networks may improve modeling of long-range dependencies and relational structure, enhancing both accuracy and interpretability - especially in tasks involving hierarchical or topological relationships. Finally, when adopting approaches that leverage knowledge derived from IA, such as the multiplication by importance scores used in EA, it is important to note that the unfrozen parts of the model may circumvent the reliance on importance masks. This is possible, for instance, because the sigmoid function never attains a value of $0.0$, and thus multiplication does not completely eliminate information associated with irrelevant image regions. In such cases, the application of these masks should be validated using causality metrics in a manner similar to that described in Appendix F.

## H   FUTURE WORK

The proposed method, in its first stage, focuses on the independent evaluation of all components in order to eliminate the influence of mutual spatial relationships on the computed patch embeddings. This is a deliberate design choice, intended to enable future independent assessment of which relationships between components are taken into account (i.e., deemed relevant) during the model's operation. Such an approach will allow for uncovering which component structures carry significance and provides a stronger foundation for validating the credibility of the model's decisions.

Additionally, while this work focuses on image data, the framework itself is modality-agnostic and generalizes to structured inputs such as sequences or graphs. In particular, less regular image representations than a grid of patches may be considered. Adaptation to other domains primarily requires the selection of appropriate architectures for processing components (ComponentNet) and global structure (ImportanceNet, ContextNet), such as the graph- and transformer-based neural networks, recurrent neural networks, or standard multi-layer perceptrons, depending on the characteristics of the input representation. Future work will investigate these extensions to further validate the framework's general-purpose applicability to interpretable learning on data with inherent structure.

## I   USE OF LLMS

We used a large language model solely for text polishing and improving clarity in the manuscript. No LLM outputs were used in any experiments, model training, or evaluation, ensuring that all results are independent of LLM assistance.

## J   LICENSES

Oxford Pets Parkhi et al. (2012) - Creative Commons Attribution-ShareAlike 4.0 International License; Stanford Cars Krause et al. (2013) - Apache License, Version 2.0; Imagenette Howard (2019) - Apache License, Version 2.0; CUB-200 Wah et al. (2022) - Apache License, Version 2.0; ImageNet Russakovsky et al. (2015) - Custom non-commercial license.

