# OpenReview forum: "What is Important? Internal Interpretability of Models Processing Data with Inherent Structure"
_ICLR.cc/2026/Conference — ICLR 2026 Conference Withdrawn Submission_

### Official Review · Reviewer_H6Gc · 2025-10-29

**Soundness:** 2
**Presentation:** 2
**Contribution:** 2
**Rating:** 2
**Confidence:** 5

**Summary:**

This paper proposes a two-stage inherently interpretable image classification framework centered on *learned importance scores*.
 (1) IA (Importance Architecture): the image is divided into non-overlapping patches, each encoded independently into embeddings (E), and an auxiliary ImportanceNet predicts an importance weight (a \in [0,1]) for each patch; the classifier aggregates the embeddings weighted by (a).
 (2) EA (Embedding Architecture): the learned (a) is frozen, and a light-weight ContextNet integrates contextual dependencies over the weighted embeddings.
 (3) PA (Pixel Architecture): the same importance mask (a) is applied at the pixel level before feeding into a standard backbone classifier.

The experiments evaluate four main aspects: semantic alignment (IoU and a custom distance metric (d)), sparsity, reproducibility, and causality tests (insertion/deletion analysis on EA). Several datasets are tested, and accuracy curves and tables are reported.

**Strengths:**

- The proposed modules make minimal architectural modifications while enforcing importance estimation *within* the inference path (i.e., not a post-hoc explanation).
- The authors visualize patch-level importance heatmaps across different patch and embedding scales (Fig. 3) and show masked images with different thresholds (t) (Figs. 6–9).

**Weaknesses:**

#### **1. Overstated novelty; limited comparison scope (biased evaluation)**

- The paper’s *Related Work* and experimental comparisons focus almost exclusively on post-hoc saliency methods (Grad-CAM, IG, SHAP, Occlusion, etc.), claiming superior semantic alignment. However, many representative in-model / inherently interpretable approaches (e.g., prototype-based, tree-structured, alignment- or concept-based models) are neither discussed nor compared. This omission systematically inflates the perceived novelty and contribution.
- Structurally, the proposed method is essentially a gating/masking mechanism followed by lightweight context integration—the paper itself states that EA fuses contextual information over the reweighted embeddings $E'$ computed by applying $a$ to $E$ .  Such “mask + context” architectures are already common in prior inherently interpretable works.
   Without fair comparison against strong in-model baselines, the claimed originality and advantage are not convincing.

------

#### **2. Simplicity and overlap with existing gating/masking paradigms**

- The IA’s main modification is a separate ImportanceNet that outputs patch-wise weights $a$, which are then applied as element-wise gates before aggregation.
   EA merely freezes $a$ and adds a shallow CNN for contextual fusion.
- Conceptually, this is equivalent to a soft attention or gating layer without additional theoretical constraint or empirical justification showing superiority over numerous prior *learned mask / attention-based interpretability* models.
   As such, the methodological novelty is limited for an ICLR-level contribution.

------

#### **3. Metric and baseline comparability issues (Sections 4.1 & 4.2)**

##### **3-a. Inconsistency in the (d) metric computation**

- The proposed $d$ metric is defined on patch-level importance values $a$ and patch-averaged semantic references (m) (L1 distance). Yet most post-hoc methods produce pixel-level saliency maps. The paper does not clearly state how these pixel-level maps were converted to patch-level scores (mean? max? normalization?). It also omits whether the same normalization and binarization thresholds were used across methods. Without consistent scaling and thresholding, cross-method comparisons on $d$  are not rigorous.
- Furthermore, the semantic mask (m) itself may be a weak reference: important patches often contain a mix of object and background pixels (e.g., a cat ear patch with more background but higher discriminative value than a belly patch).
   Hence, the (m)-based ground-truth importance may not faithfully represent semantic relevance, compromising the validity of $d$ .

##### **3-b. Incomplete accuracy baselines (Section 4.2)**

The classification accuracy comparison includes only the authors’ three variants (IA, EA, PA). This is insufficient to claim that “our interpretable mechanism preserves or improves predictive performance.” Also, clarify the hyperparameter protocol—e.g., was the PA threshold $t$ tuned only on the validation set and fixed for test reporting?

------

#### **4. Poor writing and presentation quality**

- The manuscript is difficult to read and loosely organized; key experimental details are scattered across sections.
- Most figures are non-vector graphics with tiny fonts, making them hard to interpret. Notably, Figure 5 (“Kernel density estimate plots…”) appears but is never referenced or discussed in the main text, which undermines clarity and professionalism.

**Questions:**

**Details of $d$ Computation:**
 How are the pixel-level heatmaps from different baselines unified into the patch-level $a$? Was a consistent normalization and thresholding strategy applied across all methods?

**Threshold Selection for PA:**
 In Table 2 and Figures 7–9, do the “best results” correspond to the threshold $t$ selected *only on the validation set* and then reported *once* on the test set?

**Missing Strong Baselines:**
 Why are strong *intrinsically interpretable* methods such as ProtoPNet, ProtoTree, NBDT, and B-cos not included for end-to-end comparison?

---

> ### Author Response · Authors · 2025-11-21
>
> We thank the reviewer for the detailed comments. Many of the concerns (attention-based interpretability, the use of semantic masks, and evaluation based on model accuracy) are addressed in our general response to all reviewers. Below, we address the remaining points in detail:
> - Missing strong baselines – Inherently interpretable approaches (including prototype-based, alignment- or concept-based models) are mentioned in the Related Works section. In the current revision, we have added the B-cos results, and prototype-based model results will be included in the next revision to provide a fairer comparison across approaches. In the current revision, we have also expanded the results for post-hoc methods by adding GradCAM++ and RISE.
> - Details of computation – Pixel-level saliency maps from all methods are converted to patch-level by averaging over 16×16 pixel regions, followed by normalization by the maximum value of each map. This ensures consistent scaling across all methods before computing patch-level metrics. (We added fragment describing these steps to Semanticity part.)
> - Threshold selection for PA – In Table 2, the threshold was elected based on the best validation accuracy, and the corresponding test accuracy is reported. For Figures 7 and 9, we report test accuracies directly from the full sweep (there were multiple runs for every threshold), with the bars indicating variability of accuracy.
> - Figure quality – We have replaced all relevant figures with higher-resolution images to ensure readability, including enlarging text and improving overall clarity. In addition, Figure 5 is now explicitly referenced in the main text.

---

> ### Comment · Reviewer_H6Gc · 2025-11-28
>
> Thank you for your response. The revisions have addressed some issues, but my core concerns regarding the insufficient demonstration of novelty and the weakness in baseline comparisons remain. Therefore, I am maintaining my original score.

---

### Official Review · Reviewer_qr3w · 2025-10-29

**Soundness:** 2
**Presentation:** 2
**Contribution:** 2
**Rating:** 2
**Confidence:** 3

**Summary:**

This paper proposes a framework for inherently interpretable neural networks that directly encode and quantify the importance of structured input components—such as regions in an image, tokens in a sequence, or nodes in a graph—within the model’s internal computations. The proposed approach integrates explainability through a two-stage methodology. In the first stage, convolutional networks jointly learn component representations and importance scores. In the second, a refined model relaxes structural constraints to capture spatial and contextual dependencies while preserving interpretability anchors.

**Strengths:**

The two-stage procedure elegantly separates the discovery of component importance from the modeling of global dependencies, offering clear interpretability anchors and facilitating adaptation to other data types (images, sequences, graphs).

Comprehensive experiments on multiple benchmark datasets show consistent improvements and predictive accuracy compared to state-of‑the‑art baselines.

**Weaknesses:**

Lack of novelty. Your approach analyzes data at the patch level, which seems conceptually similar to ViT‑Shapley[1]. Could you clarify what advantages your method offers compared to ViT‑Shapley, or what specific motivation drives your work beyond that prior approach?

In ViT‑Shapley, the fairness of patch‑level importance assessment is ensured through the use of Shapley value computations. How does your method guarantee comparable—or superior—fairness in evaluating the contribution of each visual patch?

Furthermore, regarding the metrics for interpretability: human studies are often considered an intuitive and widely accepted means of assessing interpretability. However, such evaluations appear absent from your paper. Could you explain the reasoning behind this choice or provide justification for omitting human studies?

[1]: Learning to Estimate Shapley Values with Vision Transformers.

**Questions:**

See Weaknesses section.

---

> ### Author Response · Authors · 2025-11-21
>
> We thank the reviewer for the detailed comments. Some of the concerns are addressed in our general response to all reviewers. Below, we address the remaining points in detail:
> - Lack of novelty – ViT-Shapley is a post-hoc explainer that trains a separate Vision Transformer to approximate Shapley values for an already trained model, leaving the original classifier unchanged and using the explainer only to interpret its predictions. In contrast, our method integrates interpretability directly into the model’s architecture. Although our solution also involves two stages in which models are trained, their purpose is entirely different. In the first stage, the IA model learns what is important for the given task. In the second stage (for which we proposed two alternative approaches, EA and PA), it leverages this knowledge, allowing us to know in advance the basis on which the model trains to make its decisions (thus a post-hoc explainer is not necessary).
> - Human studies – We did not conduct formal human studies because such studies are not easily accessible, and precisely identifying the regions that explain a class is often very challenging. Due to their nature, such evaluations are subjective, as a single object may have several alternative characteristic regions, as we have already discussed in our joint response to all reviewers. For example, in the CUB-200 dataset, one expert might focus on the color of the belly, while another might base their judgment on the shape of the beak, provided that both features allow the species to be uniquely identified. Similarly, in Stanford Cars, one person may recognize a car by the shape of the wheel arch, while someone else may focus on the appearance of the grille. In a sense, each human is an individual classifier making decisions in their own way. For this reason, we opted to search for evaluation methods that are more measurable. We evaluate importance masks quantitatively through comparisons with object masks (Section 4.1) and through insertion/deletion-inspired experiments (Section 4.2), which demonstrate that our masks indeed capture the features necessary for the model’s predictions. Moreover, we provide examples in Appendix B for qualitative inspection by the readers of our work, and we also release the code and models, enabling further assessment for arbitrary images.

---

> > ### Comment · Reviewer_qr3w · 2025-11-27
> >
> > I would like to thank the authors for their response. However, my overall impression about this paper did not change sufficiently to change the original ratings.

---

### Official Review · Reviewer_8K9A · 2025-10-31

**Soundness:** 2
**Presentation:** 2
**Contribution:** 3
**Rating:** 4
**Confidence:** 4

**Summary:**

This paper proposes a method for building intrinsically interpretable neural networks that can directly measure the importance of structured input components within the model itself, eliminating the need for post-hoc explanation techniques such as saliency maps. The approach employs a two-stage training process:
1. A specialized CNN jointly learns component-specific representations and importance scores.
2. A refined architecture with relaxed structural constraints is then fine-tuned to capture spatial dependencies and global context.

The authors evaluate the method on multiple datasets, including Oxford Pets, Stanford Cars, CUB-200, Imagenette, and ImageNet, analyzing the interpretability–performance trade-off using metrics such as semanticity, sparsity, reproducibility, and causality.
Results show that the proposed architecture achieves better semantic alignment with ground-truth annotations and higher reproducibility than traditional post-hoc saliency methods. Moreover, it provides interpretability improvements without sacrificing accuracy—and often even exceeds the predictive performance of parameter-matched baselines, both with and without pretrained backbones.

**Strengths:**

They propose a novel method that directly quantifies the importance of structured input components within the model itself, eliminating the need for post-hoc explanation techniques. The method is extensively evaluated across multiple datasets, and the quantitative results demonstrate its clear superiority over existing approaches.

**Weaknesses:**

The baseline approaches used for comparison are relatively outdated. There exist more advanced methods beyond Grad-CAM that could provide a stronger and fairer evaluation, such as RISE (Petsiuk et al., 2018) and Shap-CAM (Zheng et al., 2022). Moreover, it is unclear why the authors only compare against Grad-CAM without including its improved variant, Grad-CAM++. Limiting comparisons to older methods weakens the validity of the claimed superiority of the proposed approach.
1. Vitali Petsiuk, Abir Das, and Kate Saenko. RISE: randomized input sampling for explanation of black-box models. In British Machine Vision Conference 2018, BMVC 2018, Northumbria University, Newcastle, UK, September 3-6, 2018, page 151, 2018.
2. Quan Zheng, Ziwei Wang, Jie Zhou, and Jiwen Lu. 2022. Shap-CAM: Visual Explanations for Convolutional Neural Networks Based on Shapley Value. In Computer Vision–ECCV 2022: 17th European Conference. Springer, Tel Aviv, Israel, 459–474

**Questions:**

Could the authors provide an analysis of the running time or computational cost? This information is important, especially if the method is intended for large-scale or repeated experiments.

Additionally, it would be helpful to specify how many images were used for evaluation in each dataset.

---

> ### Author Response · Authors · 2025-11-21
>
> We thank the reviewer for the detailed comments. Some of the concerns are addressed in our general response to all reviewers. Below, we address the remaining points in detail:
> - Baseline comparison – Our approach is based on an inherently interpretable architecture, which is a novel contribution. For this reason, we initially focused on comparing it with more classical methods like Grad-CAM to establish a clear baseline. However, we agree that evaluating against newer methods is important. Accordingly, we have extended our results to include methods such as RISE and inherently interpretable B-Cos, and, as you suggested, we have also included Grad-CAM++ in our comparisons. Additionally, we note that, at least for now, we have not found a version of the code that would allow us to apply Shap-CAM method.
> - Computational cost – We measured computational time during training and inference. On a single RTX 4090 GPU, training on small datasets such as Imagenette, Cars, and Pets takes approximately 30 minutes for 100 epochs with hyperparameters set to k=128 p=16, and a batch size of 128. The average inference time per batch is 0.001686 seconds for the importance architecture (IA) and 0.002020 seconds for the embedding architecture (EA). All models are comparable in parameter count to ResNet50, and our approach remains significantly less computationally intensive than transformer-based architectures like ViT.
> - Datasets – The exact dataset splits are presented in the introduction to Section 4, dedicated to experiments.

---

### Official Review · Reviewer_37Le · 2025-10-31

**Soundness:** 2
**Presentation:** 2
**Contribution:** 2
**Rating:** 2
**Confidence:** 4

**Summary:**

The authors propose an interpretable neural network that reduces the reliance on post-hoc saliency methods. The explanation scores are learned jointly with the model parameters. The approach first quantifies the importance of individual components, after which the model learns spatial and contextual dependencies while preserving the discovered importance structure. The quality of the explanations is evaluated in terms of semanticity, sparsity, reproducibility, and causality.

**Strengths:**

- I believe the idea of quantifying component importance and preserving it is valuable, as it reduces unnecessary complexity and encourages the model to align with human-interpretable concepts.
- The methodology is tested on several datasets, demonstrating good interpretability insights.

**Weaknesses:**

- **Importance mask:** The proposed “importance mask” appears conceptually similar to an **attention mechanism**; the paper does not clearly articulate how it differs from standard attention-based interpretability.
- **The use of patches:** The claim that image patches are “semantically meaningful” is questionable—**patches are not inherently semantic units**, and true semantic meaning would require segmentation or context modeling. Moreover, the **patch-based decomposition** risks losing coherence when important concepts span multiple patches, potentially diluting concept-level importance and reducing structural interpretability.
- **Clarity of the diagram:** Figure 1 is unclear - IA, EA, and PA modules are not visually distinguished, the text is too small, and the relationships among components are ambiguous.
- **Clarity of concepts:** the interaction between **EA and PA** (whether they are trained jointly or separately) is not well explained, and the connection between the EA and its interpretability claims remains unclear.
- **Semantics:** The terminology of *“semantic structure of embeddings”* is misleading since embeddings correspond to **patches**, not true semantic entities. Moreover, measuring “semanticity” by classification accuracy on non-segmented datasets only reflects **alignment with model predictions**, not genuine semantic alignment. With respect to the metric semanticity, the paper seems to define semanticity with full object segmentation. How is semanticity defined when only part of an object is relevant? Is the goal closer to segmentation? This may explain why Grad-CAM performs well, as it typically highlights larger regions.
- **Causality metrics:** The paper’s **causality analysis** relies on insertion/deletion metrics, which do not capture causal dependencies among correlated features and are **not novel and should be cited [1].**
- **Experiments and discussion: Figure 7** contradicts the claim in Section 4.2: only CUB-200 maintains accuracy, while other datasets show degradation. The **insertion/deletion experiments lack baseline comparisons**, making it difficult to assess the actual effectiveness of the proposed method. The paper includes masked insertion/deletion images in the main text, but quantitative results are only reported in the supplementary material, and no baseline comparisons are provided.

[1] Covert, I., Lundberg, S., & Lee, S. I. (2021). Explaining by  removing: A unified framework for model explanation. Journal of Machine  Learning Research, 22(209), 1-90.

**Questions:**

- I included some questions above.

---

> ### Author Response · Authors · 2025-11-21
>
> We thank the reviewer for the detailed comments. Some of the concerns (e.g. the similarity between our method and attention-based interpretability) are addressed in our general response to all reviewers. Below, we address the remaining points in detail:
> - The use of patches – We agree that patches are not inherently semantic units. However, we do not treat them as such at any point. When we refer to the semantic nature of patches, we mean that they contain some semantically meaningful fragment relevant to the task at hand. We are not concerned about the potential risk of losing coherence when important concepts span multiple patches, because the purpose of the IA architecture is precisely to identify which patches contain something important. Even if object fragments are split at this stage, the subsequent models (EA and PA) are able to make use of them, as they take into account the spatial relationships between patches and pixels. We also considered introducing overlapping patches, but the results obtained without this modification were satisfactory. It is also worth noting that our approach generalizes to arbitrary components, provided that the semantic content of these components can be represented through embeddings in the same space.
> - Clarity of the diagram – The diagram has been replaced with a new version that we hope better conveys the idea behind our approach. Its size is constrained by the page-limit requirements, but, as we have verified, the text remains clearly readable when zoomed in.
> - Clarity of concepts – The EA and PA architectures are completely independent ways of leveraging parts of the previously trained IA model, and in particular the ImportanceNet, which is responsible for computing the importance scores for individual patches. This information is provided in the introduction to Section 3. In EA, the importance scores are applied to the patch embeddings, whereas in PA they are used to mask out non-informative image pixels. In other words, in the first stage IA learns to identify the relevant components, and in the second stage we propose two methods (EA and PA) that utilize this knowledge and are consequently internally interpretable (we know which components the models rely on when making decisions). To further clarify the relationships between the models, we have improved Figure 1. Since the interpretability of EA may indeed be questioned, we conducted an additional experiment described in Appendix F. In this experiment, we demonstrate that for the trained model there is a relationship between removing/adding low-importance patches and the model’s performance during inference. This indicates that, for the considered setup, there exists a causal relationship between the regions identified as important and the model’s decisions.
> - Semantics – We agree that the term “semantic structure of embeddings” was misleading. What we meant is simply that the fully connected layers match the embedding dimensionality, creating a bottleneck that forces the model to filter out information irrelevant to classification. We have updated the text accordingly to clarify this point.
> - Causality metrics – In the section of this response dedicated to the clarity of concepts, we partially clarified that when we refer to causality, we mean the factors on which the trained model bases its decisions (i.e., the patches indicated by the importance mask). To demonstrate this, we used an approach that was inspired by the insertion/deletion scores described in the work on the RISE method. The way these measures operate is closely related to interventions well known in causal analysis. Indeed, the approach described in [1] generalizes and further formalizes the method presented in the RISE paper. As we use it in our work, insertion-inspired approaches correspond to removing low-importance patches, whereas deletion-inspired approaches correspond to removing low-importance patches in the reverse order. We have added a citation to [1] in Appendix F.
> - Experiments and discussion – We believe that the source of the misunderstanding is that the approaches presented in Section 4.2 and in Appendix F, although both inspired by insertion/deletion scores, are entirely different. In Section 4.2, the model is trained on images whose fragments have been masked out (this is explained in detail in the section of the response common to all reviewers), whereas in Appendix F the component embeddings are zeroed out during inference (as explained above). Therefore, we consider the degradation in Figure 7 to be fully expected above a certain threshold (around 0.5) of patch importance (beyond this threshold we begin masking out image regions that are essential for recognizing its content). Thus, there is no contradiction here. The baselines in Section 4.2 are the PA(p,0.0) models, in which nothing is masked out.

---

> > ### Comment · Reviewer_37Le · 2025-11-26
> >
> > Thank you for the clarifications.
> >
> > Regarding the segmentation masks used for evaluation: although I agree that obtaining precise region-level annotations is challenging, in images where the target object (or target class) is the central focus—as is the case for most of the shown examples—evaluating explanations through their overlap with segmentation masks tends to reward methods that produce *larger* attribution maps rather than those that are most faithful to the model’s true decision process. In such cases, the IoU metric inherently favors explanations with broader spatial coverage, which may explain why methods like Grad-CAM++ achieve high IoU scores even when their attributions are not necessarily the most faithful.
> >
> > Because of this, additional analysis is needed to determine whether the high IoU scores arise from genuine model faithfulness or simply from large attribution regions.
> >
> > Your explanation makes it clear that Section 4.2 (“Accuracy”) corresponds to a different experiment. However, I still believe it is important to compare your method directly against the baseline explanation techniques previously evaluated (Grad-CAM, SHAP, IG, etc.) using occlusion- or insertion-based metrics—similar to those presented in Appendix F. Such a comparison is necessary to determine whether the improvements demonstrated by your approach translate into higher faithfulness relative to existing techniques.

---

> ### Author Response · Authors · 2025-11-27
>
> Thank you for your comment. We would like to clarify that we use a macro-averaged IoU, which evaluates whether important regions overlap with important regions and unimportant regions with unimportant ones. In this setup, overly large attribution maps would actually reduce the score, as irrelevant areas negatively affect the alignment. Metric d also accounts for this effect, mitigating the bias toward large attribution maps.
>
> Regarding your suggestion to perform an insertion/deletion analysis (similar to the analysis in Appendix F) to compare our methods directly against the baseline explanation techniques (Grad-CAM, SHAP, etc.), we are not entirely certain how such an experiment could be structured. In our approach, both the PA and EA models know from the very beginning which regions are important, because their mechanisms inherently rely (albeit in slightly different ways) on the importance masks discovered earlier in IA. In particular, for PA, described in Section 4.2 (which your suggestion refers to), the model may already have portions of the input images masked out (zeroed out) if we deem them unimportant, and thus during training it focuses exclusively on what we identify as important. Applying post-hoc explanation methods to such a model would at best reveal that the model relies on part of what we consider important, simply because the rest of the visual information has been removed by design. Conducting an insertion/deletion analysis in this context would therefore assess only the quality of post-hoc explanation of the model, rather than demonstrate that our masks themselves are faithful. We believe that the experiment in Section 4.2 already demonstrates the faithfulness of our masks: when masking out what is important, the model’s performance degrades substantially (Figure 9), whereas masking out what is unimportant does not reduce model quality (Figure 7). Could you clarify your suggestion?

---

### Author Response · Authors · 2025-11-21

We would like to express our sincere gratitude for the reviewers’ valuable and insightful comments. Since some of these remarks were, directly or indirectly, shared across multiple reviews, we would like to begin by addressing them collectively. The following section constitutes a common part of the response provided to all reviewers:
- Relation to attention-based interpretability – Our approach may indeed appear conceptually similar to attention-based interpretability. In particular, the attention coefficients play a role analogous to the elements of the importance vector a, while the values $V$ correspond to component embeddings e. For this reason, in the Related Works section we referred to methods such as the Concept Transformer and INTR. What differentiates our approach from these methods is that, in their case, the component representations $V$ are computed using the full image as input. Our intention is that these representations should not incorporate contextual information, which we aim to introduce only at a later stage of image analysis. An attention-based counterpart to our method would require training a virtual query $Q$ representing the entire image content, as well as an additional computation of the $K$ and $V$ vectors. In our view, our approach is computationally simpler and, as demonstrated through our examples, effective.
- Evaluation using semantic masks – The use of object masks in Section 4.1 raises concerns for two reasons. First, it is not obvious whether such masks constitute an appropriate reference for assessing the semanticity of an importance mask, given that only parts of an object may be relevant. Second, if importance scores are assigned to patches that may contain only fragments of an object, is such a comparison still justified? We aimed to avoid subjective assessment; therefore, we sought annotations that would allow us to evaluate the quality of the solution to the task we set. Our objective is to identify those regions of the image that are crucial for making a correct classification decision. In tasks such as animal classification, it appears natural that the reference should be the region in which the animal is visible in the image (if we can see the animal, we should be able to recognize it), and this is what we expect from the masks produced by IA. For this reason, it does not concern us that a single patch may include more than just part of the object. We agree that recognizing an object may depend only on some of its parts. We did not incorporate this aspect in our experiments for several reasons. First, it is difficult to obtain annotations that precisely indicate the regions where such discriminative parts occur (in the CUB-200 dataset only single keypoints marking the locations of bird parts are available). Second, in many cases, such part-level annotations may be questionable. One could imagine an expert recognizing a cat breed by the shape of its ears, while another relies on the pattern on its belly (we are not cat experts, so this example may be inaccurate, but it illustrates the point). Finally, the purpose of IA is to indicate the entire region that is important for the considered task, so that subsequent models can solve this task.
- Evaluation based on model accuracy – In Section 4.2 we present the performance of all models by reporting their accuracy. The purpose is, of course, to demonstrate that models relying on the components identified as important still achieve the expected results. The accuracy of IA is relatively low, as this model does not incorporate contextual information and is used solely for discovering importance masks. In contrast, the remaining models (EA and PA), which make explicit use of the identified important regions, achieve results that correspond to the state-of-the-art for, respectively, non-pretrained and pretrained models. It is worth emphasizing that PA(p,0.0) is a variant in which the mask is not used and thus, its performance serves as the baseline. To further show that the discovered importance masks highlight semantically meaningful regions, we performed an insertion/deletion-style experiment, with results in Figures 6–9. In this experiment, models are trained from scratch on data in which certain regions are occluded (unlike in conventional insertion/deletion scores, we do not mask images at inference time, but the occlusion is an integral part of the model’s operation itself). This shows that Figures 6–9 genuinely reflect the masks ability to capture importance: deleting important components lowers accuracy, while inserting unimportant ones does not. Thus, the masks are semantically relevant for the task.

Detailed responses to the reviewers’ questions and comments will be provided directly alongside the respective queries. The manuscript text has also been partially revised. Further amendments will be introduced progressively until the end of the discussion period, and the reviewers will be informed accordingly.

---

### Author Response · Authors · 2025-12-03

In this final remark, we would like to summarize the reviewers’ most important comments together with our corresponding responses. Other minor issues requiring a response (e.g., those concerning the readability of diagrams or unclear phrasing in the manuscript) have been already addressed directly in the revised version of the paper during the discussion period.

The main issues raised by the reviewers pertained to three aspects:
- Evaluation procedures – In our joint response to the reviewers, we addressed comments concerning both the use of semantic masks in Section 4.1 and the evaluation procedure based on occluding task-relevant and task-irrelevant image regions described in Section 4.2. We argue that both procedures demonstrate that the masks discovered by IA indeed identify the image regions that are informative for the task at hand. We agree that, in some cases, it may be possible to recognize the image content solely from the portions of the detected regions. However, the objective of IA is not to indicate which fragments should be used to solve the task, but rather to reveal what is in general necessary and what is not for solving it. Consequently, the regions highlighted by the importance masks enable the construction of interpretable models whose faithfulness is not a concern, as they internally rely on features that are genuinely relevant. The experiments with the PA and EA architectures indicate that this objective is indeed achieved.
- Baseline comparison – The reviewers’ comments concerned two types of missing baselines. One reviewer suggested that more recent model-explanation techniques should be used in addition to the classical ones employed in the paper. To address this concern, during the discussion stage we included results for GradCAM++ and RISE as well. Due to the absence of runnable code that would allow us to apply the Shap-CAM method, this approach has, for the time being, been omitted. The second category of comments was related to baselines involving methods that, similarly to our approach, are considered internally interpretable. In this case, we focused on methods whose interpretability involves identifying task-relevant image regions, rather than, for example, on approaches whose interpretability derives from having an inherently interpretable inference process (for this reason, we excluded NBDT). We conducted a comparison with the B-cos method. We also considered including prototype-based approaches such as PIP-Net, ProtoTree, and ProtoPNet for comparison. In practice, however, all of these methods are substantially more computationally intensive than the ResNet50-scale models considered in our study. Moreover, both ProtoTree and ProtoPNet lack readily usable validation datasets, and their implementations are generally difficult to run reliably, which precludes a fair evaluation. It is worth emphasizing that all methods added in the latest version of the manuscript achieved results inferior to those of our proposed approach, as reported in Section 4.1.
- Insufficient demonstration of novelty – We are not entirely sure how to address this comment. The differences between our method and existing approaches are described in Section 2, dedicated to related works, and concerns regarding potential similarities to attention-based interpretability were further discussed in our joint response to the reviewers. What distinguishes our approach is the method for discovering task-relevant regions in IA. This enables their validation and the construction of subsequent tools - such as PA or EA - that rely solely on this information, without the risk that the model will learn unintended correlations. Conceptually, this approach differs not only from post-hoc explanation techniques, whose validity is often questioned, but also from attention-based methods, which require computing and comparing component embeddings with respect to the entire image and are substantially more computationally expensive. In other words, we are able to obtain a similar outcome in a considerably simpler manner. Moreover, the proposed approach has the additional advantage of enabling a decoupled analysis of the importance of individual components and the relations between them.

In conclusion, we would once again like to thank the reviewers for their time and valuable comments. They have helped us prepare an improved version of our work.

---

### Note · Authors · 2025-12-03

**Comment:**

After internal discussions among the authors, we have decided to withdraw our submission. We would like to once again thank the reviewers and the conference organizers for their work. Based on their feedback, we plan to prepare an improved version of the manuscript in which we will reconsider the possibility of adding arguments and experiments that may more effectively convince the reader of our approach.

**Withdrawal Confirmation:**

I have read and agree with the venue's withdrawal policy on behalf of myself and my co-authors.